# Testing and evaluation of a new airborne system for continuous $N_2O$, $CO_2$, CO, and $H_2O$ measurements: the Frequent Calibration High-performance Airborne Observation System (FCHAOS)

Alexander Gvakharia[1], Eric A. Kort[1], Mackenzie L. Smith[1,2], and Stephen Conley[2]

[1]Climate and Space Sciences and Engineering, University of Michigan, Ann Arbor, Michigan, USA
[2]Scientific Aviation, Boulder, Colorado, USA

*Correspondence to:* Alexander Gvakharia (agvak@umich.edu)

**Abstract.** We present the development and assessment of a new flight system that uses a commercially available continuous-wave, tunable infrared laser direct absorption spectrometer to measure $N_2O$, $CO_2$, CO, and $H_2O$. When the commercial system is operated in an off-the-shelf manner, we find a clear cabin pressure/altitude dependency for $N_2O$, $CO_2$, and CO. The characteristics of this artifact make it difficult to reconcile with conventional calibration methods. We present a novel procedure that extends upon traditional calibration approaches in a high-flow system with high-frequency, short-duration sampling of a known calibration gas of near-ambient concentration. This approach corrects for cabin pressure dependency as well as other sources of drift in the analyzer while maintaining a ~90% duty cycle for 1 Hz sampling. Assessment and validation of the flight system with both extensive in-flight calibrations and comparisons with other flight-proven sensors demonstrate the validity of this method. In-flight $1\sigma$ precision is estimated at 0.05 ppb, 0.10 ppm, 1.00 ppb, and 10 ppm for $N_2O$, $CO_2$, CO, and $H_2O$ respectively, and traceability to WMO standards ($1\sigma$) is 0.28 ppb, 0.33 ppm, and 1.92 ppb for $N_2O$, $CO_2$, and CO. We show the system is capable of precise, accurate 1 Hz airborne observations of $N_2O$, $CO_2$, CO, and $H_2O$ and highlight flight data illustrating the value of this analyzer for studying $N_2O$ emissions on ~100 km spatial scales.

## 1 Introduction

Continuous, 1 Hz airborne observations of atmospheric greenhouse gases and pollutants provide essential information for direct quantification of emissions (Karion et al., 2015; Peischl et al., 2015; Smith et al., 2015; Kort et al., 2016), assessment of modeled representations of emissions and transport (Wofsy, 2011; O'Shea et al., 2014), and validation of remote sensing observations (Tanaka et al., 2016; Inoue et al., 2016; Frankenberg et al., 2016). Advances in the last decade have facilitated widespread, high-precision, high-accuracy continuous airborne observations of $CH_4$, $CO_2$, CO, and $H_2O$ (Chen et al., 2010; Karion et al., 2013; Filges et al., 2015). These observations have proven particularly valuable for quantifying emissions from individual, large emitting point sources (Conley et al., 2017; Mehrotra et al., 2017) as well as constraining emissions of highly heterogeneous processes on 10-100 km scales (Karion et al., 2015; Peischl et al., 2015; Smith et al., 2015; Kort et al., 2016). Continuous, 1 Hz airborne sampling of $N_2O$ with high accuracy and precision has proven more elusive, with limited aircraft campaigns reporting continuous airborne $N_2O$ (Kort et al., 2011; Wofsy, 2011; Xiang et al., 2013), systems being large and

challenging to operate with frequent attention to supplies of cryogens (Santoni et al., 2014), and newer systems showing large cabin pressure dependencies (Pitt et al., 2016).

$N_2O$ is a potent greenhouse gas with natural and anthropogenic sources, and is currently the single most impactful anthropogenic ozone-depleting substance actively emitted to the atmosphere (Ravishankara et al., 2009). Atmospheric emissions of $N_2O$ have been steadily rising over time (Myhre et al., 2013), but attempts to better quantify, understand, and constrain anthropogenic emissions have been hindered by high uncertainties in model estimates and limited observational constraints (Ciais et al., 2013; Davidson and Kanter, 2014). The poor understanding of $N_2O$ emissions processes is attributable to a combination of high spatial and temporal variability (Monni et al., 2007) that is hard to observe and represent, and a lack of direct observational data of emissions sources (Brown et al., 2001). The largest source of anthropogenic $N_2O$, contributing 66% of global $N_2O$ emissions, is agricultural activity (Davidson and Kanter, 2014). Some of these emissions are a direct product of human activity, such as the fertilizer production process, which has grown to $100 \, \text{Tg N yr}^{-1}$ since the development of the Haber-Bosch process in 1908 (Erisman et al., 2008). Other anthropogenic emissions, such as from applied fertilizer, are harder to observe and represent as environmental factors including soil moisture, temperature, and crop type all influence emissions (Dalal et al., 2003; Ruser et al., 2006; Griffis et al., 2017).

A diverse range of approaches have been utilized in attempts to measure $N_2O$ emissions (Denmead, 2008; Rapson and Dacres, 2014). Flux chambers can quantify emissions from areas on the order of square meters (Bouwman et al., 2002; Marinho et al., 2004; Turner et al., 2008; Chadwick et al., 2014). Given the heterogeneity in $N_2O$ emission processes, extrapolation of limited flux chambers to accurately represent domains on the orders of 10-100 square km remains challenging (Pennock et al., 2005; Flechard et al., 2007). The eddy covariance approach deploys sensors on towers to estimate fluxes on a 1-10 $\text{km}^2$ scale (Dalal et al., 2003; Pattey et al., 2007), but not beyond that range, thus encountering similar representation challenges as flux chambers. Bottom-up modeling of emissions processes (Del Grosso et al., 2006; Tian et al., 2015) can represent emissions at a range of scales. The models are typically trained and evaluated with data from flux-chambers and then simulate emissions at a continental to global scale. Evaluation of these representations then can happen at the larger scales, where top-down atmospheric inversions (Kort et al., 2008, 2011; Miller et al., 2012; Thompson et al., 2014; Chen et al., 2016; Griffis et al., 2017; Nevison et al., 2018) have challenged modeled and inventoried emissions and often found large discrepancies exceeding 100% (Miller et al., 2012). To better understand these divergences as well as to properly assess the representation of flux-chamber and eddy covariance measurements, we need observational constraints at 10-100 square km spatial scales.

Continuous, 1 Hz airborne measurements can provide information at this critical spatial scale, in addition to providing observational constraints for large point sources ($N_2O$ fertilizer production facilities present a potentially important source of $N_2O$ emissions). To get good, useful data, aircraft studies require instruments that have high precision, a fast response time, and are relatively robust to changes in the environment (Fried et al., 2008). Continuous-wave tunable infrared laser direct absorption spectrometers (CW-TILDAS) can satisfy those requirements and are an appropriate choice for airborne instrumentation (Rannik et al., 2015).

Infrared laser spectrometers have been widely used in airborne studies. They often employ an in-flight calibration to correct for spectral drift that can occur over several hours of measurement (O'Shea et al., 2013; Santoni et al., 2014). Zero air with no

gases in the absorption spectrum can also be used to adjust the spectral baseline for more accurate measurements, particularly if the desired gas has a weak absorption feature (Yacovitch et al., 2014; Smith et al., 2015). One recent study to measure $N_2O$ emissions with such an instrument reported their assessment of its performance and found artifacts in the data primarily due to changes in airplane cabin pressure (Pitt et al., 2016), significantly impacting the duty cycle of the analyzer and its utility during vertical profiles. To deploy a CW-TILDAS for $N_2O$ observation as in Pitt et al. (2016), problems can arise if drifts occur on a timescale faster than the conventional calibration period of 0.5-1 hour. Also, at low flow rates (0.1 - 1 slpm), $N_2O$ can take a long time to equilibrate, and this can have a negative impact on the instrument's duty cycle (Santoni et al., 2014). The efficacy of airborne instrumentation for $N_2O$ measurements would benefit from improvements to such limitations.

We present the Frequent Calibration High-performance Airborne Observation System (FCHAOS), utilizing a TILDAS instrument and an updated calibration technique, to make $N_2O$ measurements that can be utilized for calculating facility emissions, mass balance fluxes, and regional inversions. Rather than relying on spectral zeros and infrequent in-flight calibrations to correct for drift on large time-scales, we use short, frequent calibration measurements to resolve both long-term spectral drift and short-term environmental effects. This research was part of the Fertilizer Emissions Airborne Study (FEAST) campaign in spring 2017 targeting $N_2O$ and other greenhouse gas emissions in the southern Mississippi River Valley region of the USA. In this manuscript we discuss the operation and set-up of the instrumentation involved in the airborne flight system. We discuss test flights done to assess the off-the-shelf operation and the associated flaws. We then present our solution to improve instrument performance with short, frequent calibrations and validation by in-flight calibrations and comparison with a flight-proven Picarro cavity ring-down spectrometer.

## 2 Instrumentation

### 2.1 CW-TILDAS description

The core of our system is an Aerodyne mini spectrometer. The spectrometer uses a mid-IR, continuous-wave, distributed feedback laser with a frequency of 2227 cm$^{-1}$ (nanoplus, Germany). The laser is mounted on a copper Peltier device which keeps the laser temperature stable at ~17 °C and is regulated by a thermoelectric chiller held at 23 °C (Oasis 3, Solid State Cooling, USA). This laser is optically aligned into a 0.5 L astigmatic mirror multipass absorption Herriott cell (McManus et al., 1995). The refraction pattern in the cell is optimized to produce a total path length of 76 meters before the beam exits the cell and is aligned into a photodetector. The cell itself is sealed and held at ~40 Torr. The space outside of the cell is subject to variations in external pressure. The laser's output frequency can be adjusted by ramping the current, sweeping across a frequency range of approximately 2227.4-2227.9 cm$^{-1}$. This range contains transition lines for $H_2O$, $CO_2$, CO, and $N_2O$, allowing the photodetector to measure the laser transmission intensity at each of these transitions (Nelson et al., 2002).

The mole fractions of $N_2O$, $CO_2$, CO, and $H_2O$ are reported using the TDLWintel software as described in Nelson et al. (2002) and Nelson et al. (2004). The retrieval uses the Beer-Lambert law, where the absorption intensity, path length, and molar absorptivity enable calculation of gas mixing ratio. The absorption spectrum is fit in real-time with a Voigt density profile using the Levenberg-Marquardt algorithm, allowing retrievals at 1 Hz (Nelson et al., 2004). The exact frequencies of

the line transitions and absorption cross-sections are obtained from the HITRAN2012 database (Rothman et al., 2013). Pressure and temperature data acquired from sensors in the cell are used to account for broadening effects in the fit.

## 2.2 Set-up and payload

We integrated the FCHAOS system on a single-engine Mooney M20R aircraft from Scientific Aviation. Figure 1 shows the flow diagram for our system. The inlet line to the instrument is ~5 m PVDF Kynar tubing. The inlet line is rear-facing on the right wing to reduce liquid and particle contamination of the line, with the plane exhaust located on the left wing, minimizing exhaust contamination. A membrane disc filter (Pall, USA) is also used to block particulates from entering the cell. Using a mass flow controller (MC-5SLPM-D, Alicat Scientific, USA), we set a flow rate of 1.5 slpm. The MFC is placed downstream of the filter to prevent damage due to rogue particulates. The instrument cell is pressurized on the ground to 40 Torr using a dry scroll pump (IDP-3, Agilent Technologies, USA) and a needle valve (SS-1RS4, Swagelok, USA) directly upstream of the pump for adjusting the target pressure given a defined mass flow rate. The use of mass flow control enables rapid switching between calibration gas and ambient air without inducing pressure fluctuations or ringing in the cell. The mass flow control setup is a closed system (no excess flow), thus ensuring no contamination of other inlets and minimal waste of calibration gas. Pressure-control systems that are optimally tuned may achieve similar performance, but even with an excess flow to reduce pressure pulses, it is difficult to reach similar performance as with mass flow control. Figure 2 illustrates respective performance in-flight of a pressure and mass flow control configuration for our instrument. Two 2 L aluminum carbon-fiber-wrapped compressed air cylinders are securely strapped in the plane. These tanks are outfitted with stainless steel regulators (51-14B-590, Air Liquide, USA) and stored calibration gases. Two three-way solenoid valves (009-0294-900, Parker-Hannifin, USA) control the air flow between the tanks and the inlet line.

The additional payload is set up on the Mooney as described in Conley et al. (2014) and Conley et al. (2017). Temperature and relative humidity are recorded with a humidity probe (HMP60, Vaisala, Finland). A cavity ring-down spectrometer (G2301-f, Picarro, USA) measures $CH_4$, $CO_2$, and $H_2O$ as described in Crosson (2008). Ozone is measured with an ozone monitor (Model 202, 2B Technologies, USA). Wind speed and direction are calculated using a differential GPS method as in Conley et al. (2014). The Mooney aircraft is not pressurized, so the instrument experiences pressure variation as the aircraft profiles.

Lag time between when air enters an instrument's inlet line and when it is measured in the cell is determined by breathing close to the inlet tube and recording sharp rises in $CO_2$ and $H_2O$ mixing ratios. For FEAST lag times were measured at 3 s and 5 s for the FCHAOS and Picarro G2301-f respectively, values confirmed in flight by comparing variability with temperature and RH data from the humidity probe. These lag times are used in post-processing to match avionics and GPS data with the co-located molar ratios from the FCHAOS and G2301-f. Though lag times will vary with altitude, given the flow-rates, inlet line volumes, and altitude range of the Mooney aircraft it is essentially constant for the data presented in this manuscript.

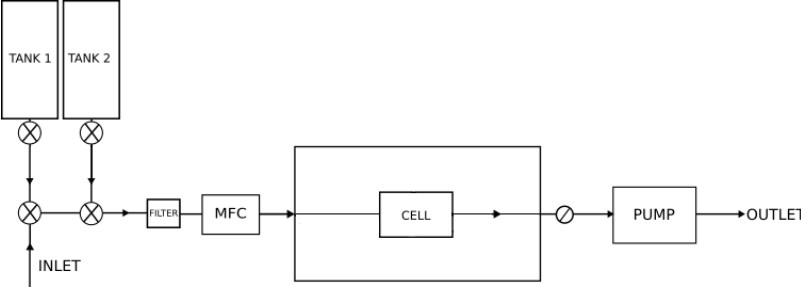

**Figure 1.** Schematic of FCHAOS, where air flows from the inlet line through the solenoid valves, past the filter to the mass flow controller (MFC), through the instrument cell, a needle valve, and finally the vacuum pump. When calibrating the solenoid valves are actuated to direct flow from each individual calibration tank into the cell directly.

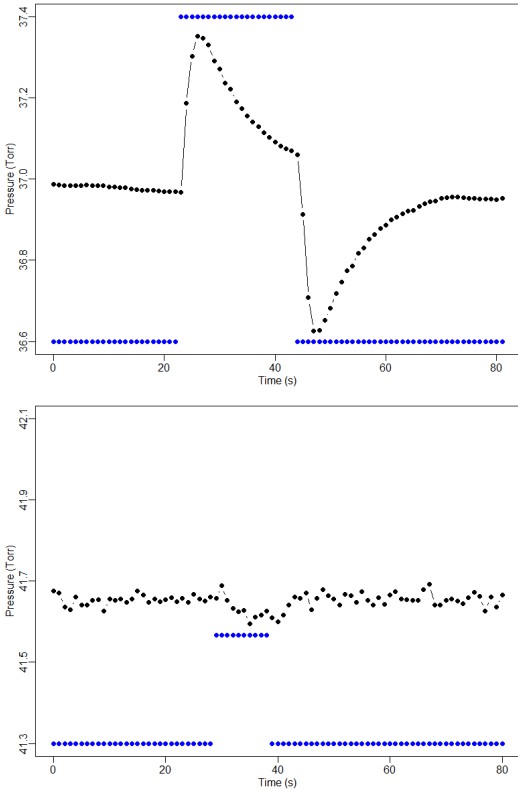

**Figure 2.** Cell pressure (black) in response to actuating a solenoid and sampling a standard cylinder (blue indicates solenoid position). The pressure control setup (top panel), including excess flow, exhibits significant pressure perturbations and residual transients that persist longer than desired calibration time. The mass flow control setup (bottom panel) shows pressure perturbations of shorter duration and on the order of 0.04 Torr, 20 times smaller than with pressure control.

## 2.3 Calibration

We performed pre-flight calibrations on the ground for both the FCHAOS and Picarro G2301-f using two air cylinders calibrated to a NOAA WMO greenhouse gas scale (X2007, X2004A, X2014A, X2006A for $CO_2$, $CH_4$, CO, and $N_2O$ respectively) (WMO, 2015). Both cylinders had mixing ratios of $CO_2$ (Zhao et al., 1997; Zhao and Tans, 2006; Tans et al., 2017), $CH_4$ (Dlu-

gokencky et al., 2005), CO (Novelli et al., 2003), and $N_2O$ (Hall et al., 2007) near ambient atmospheric levels, with one as a high-span standard and the other as a low-span.

We sequentially sampled these cylinders for multiple cycles, and compared the measured mixing ratios for each gas to the reported value on the WMO scale. We consider known values $X_{true}$ against the measured values $X_{measured}$, and a linear fit provides the slope $m$ and intercept $b$ such that $X_{true} = m*X_{measured} + b$.

We filled the two in-flight calibration tanks used with the FCHAOS for FEAST with a separate custom mixture that contained atmospheric levels of $N_2O$, $CO_2$, and CO. We calibrated the mixing ratios using the WMO standard cylinders by sampling the target cylinders in between the WMO standards. During flights, we used one tank as a single-point calibration gas, while the other was used as a check gas to assess the instrument's traceability. We elaborate on these processes in Sect. 3.2 and 4.1.

We assessed the stability in slope of the instrument by performing calibrations separated by months before and after the

FEAST campaign. Over the course of four months, the slopes for $N_2O$, $CO_2$, and CO changed by 0.4%, 0.01%, and 0.5%. The impact of any variation in slope depends on the difference between ambient levels and calibration gas values. For the operation of FCHAOS, we use calibration gas with mixing ratios near ambient levels. Typical atmospheric ambient levels of $N_2O$ are ~335 ppb, so with a calibration gas at ~330 ppb, the long-term variation due to linearity is 0.4% of 5 ppb, or 0.02 ppb, an uncertainty that is within our 1 Hz precision as reported in Sect. 4.1. For $CO_2$ and CO, which have ambient atmospheric levels

of ~400 ppm and ~155 ppb, we use calibration gases with ~390 ppm and ~150 ppb, and the impacts due to variation in slope are 0.01 ppm and 0.025 ppb, respectively. If zero air were used instead, the impact on $N_2O$ would be on the order of 0.4% of 335 ppb, or 1.3 ppb, an order of magnitude larger, with similar impacts for $CO_2$ and CO. By using calibration gases close to ambient levels we eliminate our sensitivity to drift in the instrument's slope and thus can a single gas target for in-flight calibration to correct only for intercept variability.

## 2.4 Water vapor

Spectroscopic measurements of atmospheric species are sensitive to dilution and broadening effects due to water vapor (Chen et al., 2010, 2013; Rella et al., 2013). TDLWintel, in its retrieval algorithm, corrects for water dilution and uses $H_2O$ broadening coefficients to mitigate the effect of water vapor on the spectral lines, directly reporting dry molar fractions for $N_2O$, $CO_2$, and CO (Lebegue et al., 2016; Pitt et al., 2016). This coefficient is the ratio of spectral line broadening due to water pressure

compared to air pressure broadening. To determine the coefficients, we conducted a test where dry tank air was sampled with varying amounts of water vapor. We used a similar approach as in Lebegue et al. (2016). We used a moist filter along with variable flow through parallel dry tubing, enabling some control of the water vapor content by modulating the relative flows over the moist filter compared to the dry tubing. We sampled at varying humidity starting at ~1.6% $H_2O$ and decreasing to near

0, spanning a typical range of atmospheric water vapor. Using spectral playback in TDLWintel, we were able to re-analyze the spectra with various broadening coefficients until we found the optimum values as in Pitt et al. (2016). Figure 3 shows the measurement data from the test using our optimized broadening coefficients of 1.33, 1.93, and 1 for $N_2O$, $CO_2$, and CO, respectively. The dry value is determined from prolonged sampling of dry air only from the standard tank. The deviation from this is shown as a function of water vapor. The gray line shows a moving average with a 10 s window. The RMS difference in $N_2O$, $CO_2$, and CO was 0.023 ppb, 0.076 ppm, and 0.75 ppb, respectively. These are used as the uncertainty in water vapor correction, as in Pitt et al. (2016). For CO, a coefficient of 1 corresponds to purely a dilution correction. Larger values of the coefficient do not improve the dependency. As highlighted by Pitt et al. (2016), water broadening coefficients must be determined by users for their own instrument as these can vary for each analyzer and can introduce substantial errors in correcting to dry air mole fraction.

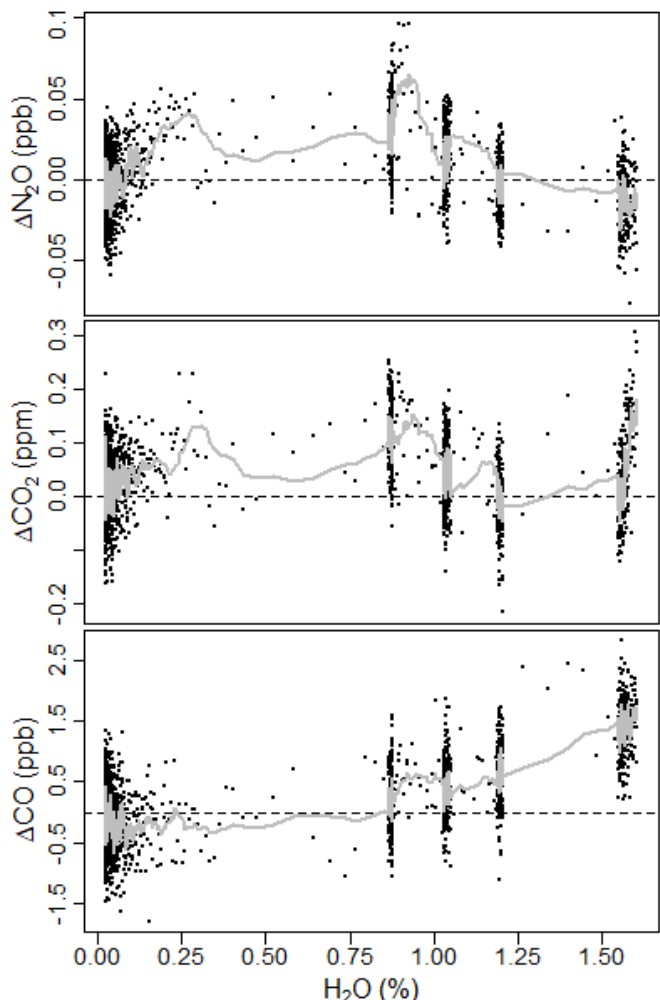

**Figure 3.** Residual uncertainty in Water vapor correction for $N_2O$, $CO_2$, and CO with broadening coefficients of 1.33, 1.93, and 1, respectively. Black dots are the deviation from the dry value, with a moving average (10 s) depicted in gray.

## 3 In-flight operation

### 3.1 Null Test

For an instrument to be well-suited for airborne observation, resistance to environmental effects is paramount. A "null test," where an instrument samples air with known mixing ratios in flight while subject to variation in cabin pressure, air temperature, etc., can be useful in evaluating its robustness as shown in Chen et al. (2010) and Karion et al. (2013). We conducted two null tests using the FCHAOS, once during a test flight in Colorado, once during a research flight in our target region in the lower Mississippi River basin. Figure 4 shows $N_2O$, $CO_2$, and CO mixing ratios observed by the FCHAOS while sampling tank air

during a vertical profile descent. As the altitude decreases, there is a clear dependence due to the cabin pressure changing similar to what was reported in Pitt et al. (2016). As mentioned in Sect. 2, though the cell is pressurized, the rest of the instrument is not, and since the aircraft cabin is not pressurized, our system thus experiences any change in ambient pressure. Correcting or mitigating this cabin pressure artifact is necessary for FCHAOS to be capable of accurate airborne in-situ sampling.

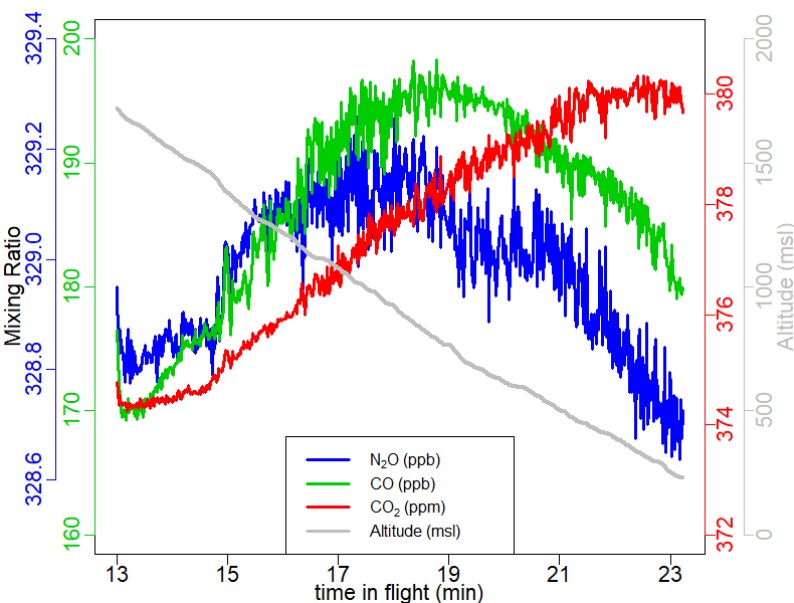

**Figure 4.** A null test demonstrates artifacts when operating the instrument in an off-the-shelf manner. Drift occurs in $N_2O$, $CO_2$, and CO due to changes in cabin pressure that occur with changing aircraft altitude.

## 3.2 Frequent calibration correction

The cause of the cabin pressure dependence is not immediately evident. One possible explanation could be an optical fringe pattern in the absorption spectrum that moves with changing cabin pressure. Acceleration during altitude change could also create g-force or electrical (via engine surges) changes that propagate through the instrument system. Without needing to pinpoint the cause, we know the time period of the artifact presents on the order of many minutes, with a typical aircraft climb rate of 500 ft/min. Thus a correction that occurs on a shorter time spacing could remedy the drift. To account for both spectral drift in the instrument that occurs on the order of hours and cabin pressure-related artifacts that emerge on the order of minutes, we developed an empirical correction procedure using frequent measurements of a calibration gas.

The procedure is as follows. Every 2 min, we actuate the solenoid valve to sample tank air for 10 s. We determined the calibration frequency of 2 min through a sensitivity test using null test data. By adjusting the calibration frequency and measuring the precision, we found similar $1\sigma$ uncertainties at 1 min and 2 min frequencies, but an increase in uncertainty at 4 min and beyond, making 2 min good for reducing gas consumption while maintaining high precision. We allow 5 s of flush time, leaving 5 s of measurement time. We determined the flush time duration of 5 s by sampling tank air in a lab setting at the same

flow-rate and cell pressure as in-flight operation and measuring equilibration time. We calculate the average measured mole fraction of $N_2O$, $CO_2$, and CO in these 5 s. Figure 5 demonstrates a typical in-flight calibration.

For each species we then interpolated in time using a Forsythe, Malcom, and Moler cubic spline between each measured calibration gas value and subtracted the known "true" value from this interpolation, giving us correction as a function of time.

5   We then subtract this calibration curve from the raw data. Figure 6 shows both raw $CO_2$ data and the correction we derive using the frequent calibration method from one of our flights. As mentioned above, the gas was on for 10 s, along with 5 s of post-calibration time removed to account for equilibration back to ambient sampling, resulting in a loss of 15 s of atmospheric observations every 120 s for an 87.5% duty cycle. As mentioned in Sect. 2.3, the calibration cylinder mixing ratios are near atmospheric values. As seen in Santoni et al. (2014), $N_2O$ can take a long time to equilibrate between measurement sources

10   due to its propensity to stick to tubing. Thus, choosing calibration values close to ambient is critical for maintaining short flush times. This also holds for $CO_2$, though less so for CO. Artifacts that occur on shorter time-frames, such as those induced by a short-duration turbulence event, will not be corrected with this method.

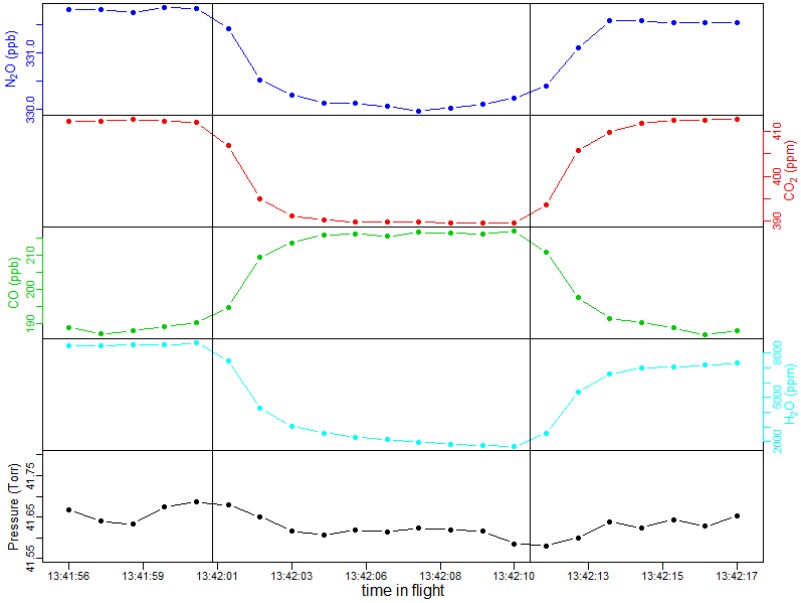

**Figure 5.** Example of in-flight calibration, showing time series of $N_2O$, $CO_2$, CO, $H_2O$, and cell pressure. Vertical lines indicate when the solenoid valve was actuated or closed. The first 5 seconds of each calibration are treated as equilibration time, and the last 5 seconds are used to find a mean calibration value.

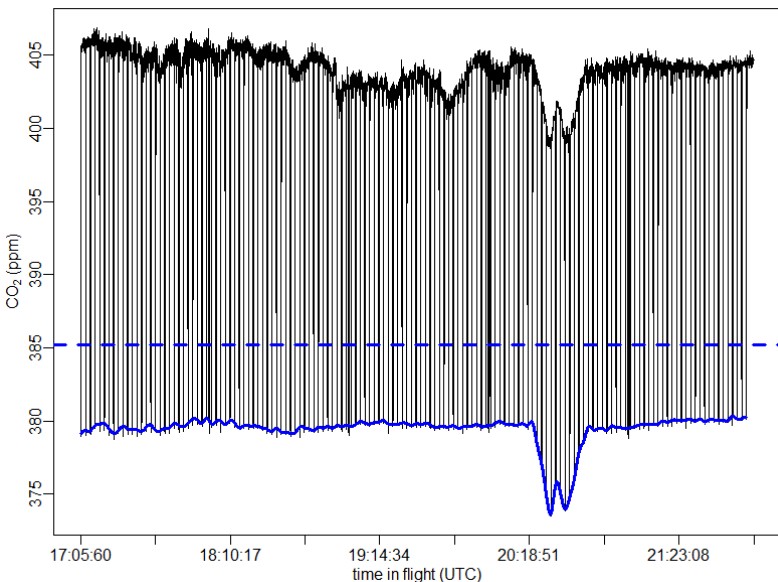

**Figure 6.** Raw $CO_2$ (black) measured by the FCHAOS for an entire flight, with frequent low dips due to calibrations. The blue dashed line indicates the "true" value of the calibration gas, the blue solid line shows the calibration curve obtained by interpolating between each calibration instance. The difference between the dashed and solid blue lines is used to correct for drift.

## 4    Calibration results and comparison with Picarro G2301-f

Figure 7 shows measurements from two null tests, one on April 26, 2017 in Colorado and one on May 2, 2017 in the Mississippi River Valley, the same null test as from Fig. 4. For each null test, the figure shows both the raw $N_2O$, CO, and $CO_2$ measurements and the corrected data following our calibration, along with the aircraft altitude. Our calibration method accounts for
the clear cabin pressure/altitude dependence. During a null test FCHAOS samples tank air uninterrupted, rather than making a calibration measurement every 2 min as in the frequent calibration procedure described in Sect. 3.2. Thus, we average 5 s of data from every 120 s to simulate the normal operation mode. Even after correction there is some residual coherent variability evident at the 15 min mark of the null test shown in the bottom 2 rows of Fig. 7, but this potential feature remains still within our 1 Hz precision.

Given the repeatable, smooth nature of the cabin pressure artifact, it would seem possible to use just the cabin pressure data to empirically correct for the artifact, without running frequent calibrations. This method would not account for long-term spectral drift however or traceability, and relies on the assumption that the cabin pressure artifact will be stable and repeatable. These weaknesses compromise such an approach.

Figure 8 compares the raw $CO_2$ data from the Picarro G2301-f and FCHAOS during a research flight along with altitude,
and a second comparison once the FCHAOS data is corrected. The difference between the two instruments is shown in the top

2 panels. The most significant discrepancies occur during the vertical profile section of the flight. Following calibration, the deviation during profiling is eliminated, and the $1\sigma$ uncertainty in the difference is reduced from 1.15 ppm to 0.28 ppm.

For the FEAST campaign, in post-processing it became evident that a persistent offset of 0.51 ppm existed for $CO_2$ between the Picarro and FCHAOS. For the $CO_2$ comparisons in this manuscript, we have corrected for this bias. We believe the origin of this offset to be related to regulator contamination of a calibration gas cylinder and/or tubing used in conjunction with the regulator. With subsequent investigation it has been difficult to identify the exact cause. We do note that in comparing the Picarro and FCHAOS instruments, they both are calibrated with dry tank air, whereas the in-flight comparison is while measuring wet ambient air. Any residual water vapor sensitivity not corrected for either analyzer can manifest as an apparent bias, and this further emphasizes the need to validate water vapor corrections, as pointed out by Pitt et al. (2016), and further outlined for FCHAOS above in the discussion of the water vapor correction.

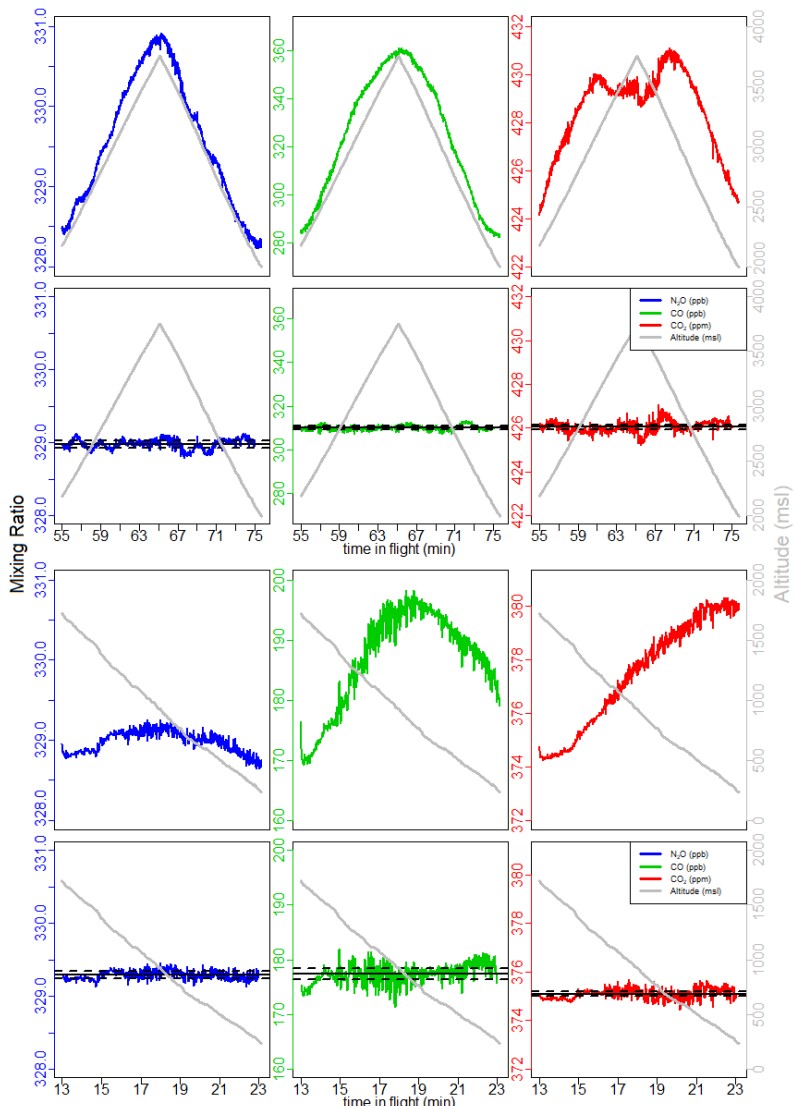

**Figure 7.** Top two rows show FCHAOS data from a null test on April 26, 2017, bottom two rows shows data from a null test on May 2, 2017, the same seen in Fig. 4. Rows 1 and 3 show $N_2O$, CO, and $CO_2$ during the null test before any calibration, rows 2 and 4 show the gas data following the frequent calibration correction. The procedure removes cabin pressure dependence and calibrates for linear drift. Black horizontal lines show mean and $1\sigma$ uncertainty.

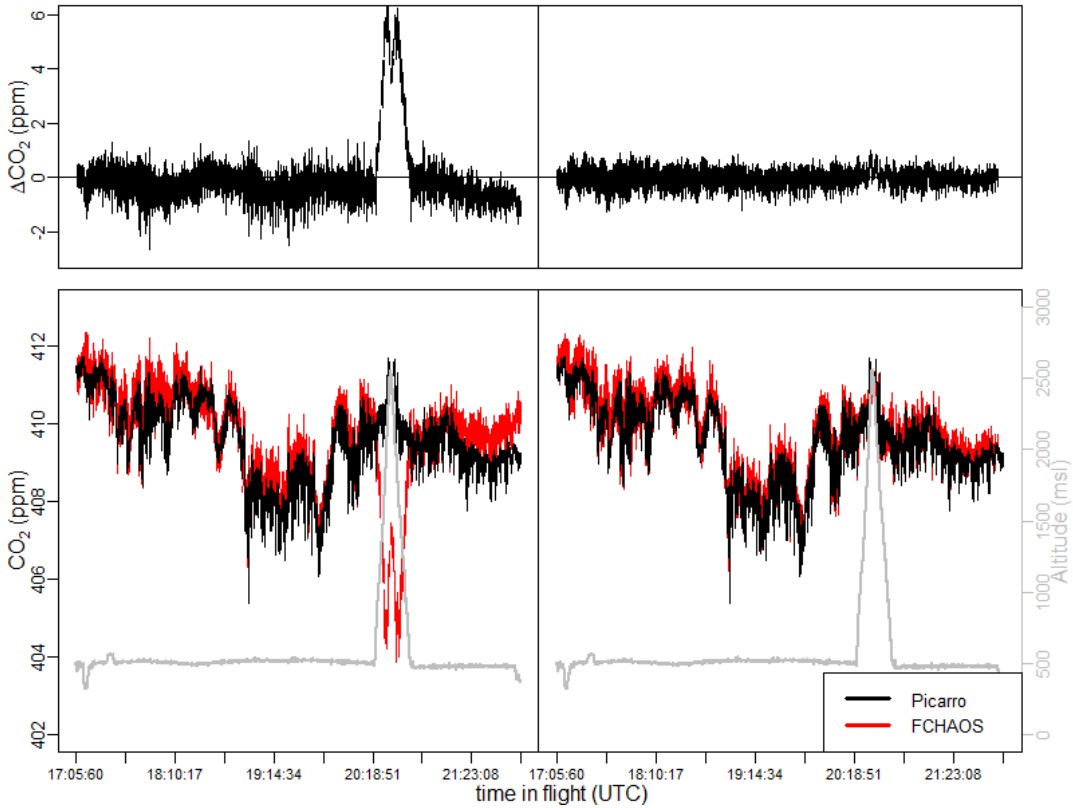

**Figure 8.** Bottom panels show Picarro G2301-f and uncalibrated FCHAOS $CO_2$ time series on left, Picarro G2301-f and calibrated FCHAOS $CO_2$ on right. Top panels show difference between the two instruments with and without FCHAOS calibration. The calibration procedure corrects for any artifacts in the FCHAOS data correlated with aircraft altitude.

The raw $H_2O$ measurements exhibit good agreement between FCHAOS and the Picarro G2301-f. The $H_2O$ data was not calibrated or adjusted in any way, as there appeared to be small impact from cabin pressure variance and it is not well characterized. Figure 9 shows a histogram of the differences in FCHAOS and Picarro G2301-f $H_2O$ and $CO_2$ (following calibration) for ~40 hours of research flight time. Figure 10 shows the differences as a function of time for all flight data. For $H_2O$, we find
5  a mean difference between the two instruments of 180 ppm, a median of 180 ppm, and $1\sigma$ of 340 ppm, shown in the figures as solid and dashed lines. In-flight $1\sigma$ precision for $H_2O$ from the Picarro G2301-f has been reported as 100 ppm (Crosson, 2008), while the in-flight $1\sigma$ precision for the FCHAOS was found to be 10 ppm.

Why does water vapor not exhibit the same sensitivities as the other gases? To assess the sensitivity for water vapor to cabin pressure is more challenging given the long equilibration time. In Fig. 11 we show $H_2O$ during the null test. On the null
10  test where water vapor has previously equilibrated, some altitude-dependent sensitivity is apparent (~60 ppm). Our calibration approach cannot well address this potential residual sensitivity given the long equilibration time required for $H_2O$. Does this potential artifact matter? In comparison with the Picarro analyzer (Fig. 11) we see no evident residual sensitivity to altitude. Given relative uncertainties, we cannot eliminate the presence of a vertical sensitivity of 10s ppm for water vapor.

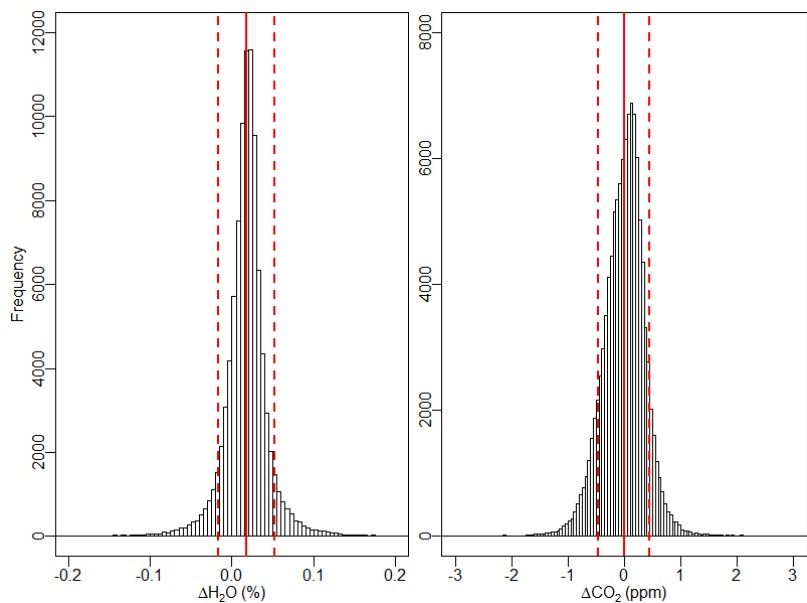

**Figure 9.** Histogram of difference between $H_2O$ and $CO_2$ mixing ratios from FCHAOS and the Picarro G2301-f. FCHAOS $CO_2$ has been calibrated, while $H_2O$ has not. For $H_2O$, mean of 0.018% or 180 ppm, median of 0.018% or 180 ppm, $1\sigma$ of 0.034% or 340 ppm, where Picarro G2301-f precision is 100 ppm. For $CO_2$, mean of 0 ppm, median of 0.024 ppm, $1\sigma$ of 0.45 ppm.

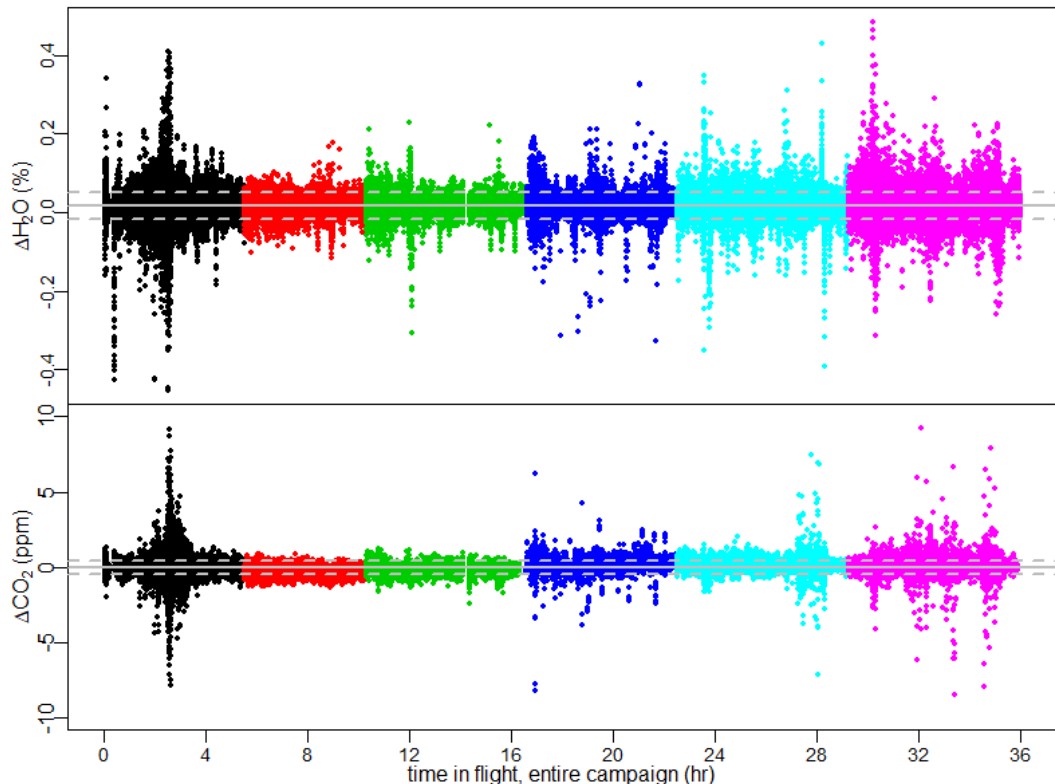

**Figure 10.** Difference as function of flight time for FCHAOS and Picarro G2301-f $H_2O$ and $CO_2$ for all research flights. Colors separate flight days, gray lines indicate mean and $1\sigma$ uncertainty. Largest deviations occur when sampling in the immediate near-field of large point sources where some mismatched lag times contribute to deviations.

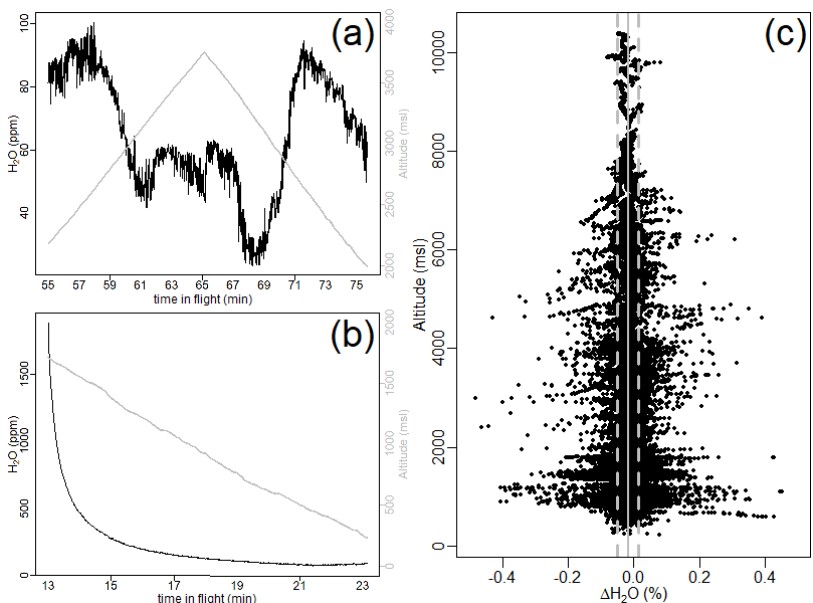

**Figure 11.** Panels (a) and (b) show $H_2O$ during null tests from Fig. 7. In panel (b) $H_2O$ hasn't fully equilibrated. In panel (a), $H_2O$ previously equilibrated and there does appear to be a dependence on altitude on the order of 60 ppm. As seen in panel (c), the difference in $H_2O$ between the Picarro and FCHAOS over the entire campaign does not exhibit an altitude dependence, so while there may be some altitude sensitivity, the effect is relatively small compared to typical atmospheric concentrations of $H_2O$ and our overall water vapor uncertainty.

### 4.1 Precision and Accuracy

To assess the FCHAOS precision, we consider flight data during a null test when the altitude did not significantly change. We find 1 s precisions of $\pm 0.05$ ppb, $\pm 0.10$ ppm, $\pm 1.00$ ppb, and $\pm 10$ ppm for $N_2O$, $CO_2$, CO, and $H_2O$ respectively. This is about a factor of 2 greater than the performance on the ground in a lab setting, with $1\sigma$ precisions of 0.02 ppb, 0.05 ppm, 0.50 ppb, and 7 ppm. Considering an Allan variance analysis of both the in-flight null test and in-lab study, the same result holds, in that the Allan variance in the air closely matches the ground, with performance degraded by a factor of 2.

In addition to the frequent calibrations every two minutes, a second cylinder is sampled every hour for 25 s as a "check gas" to test the traceability of the in-flight system. The last 5 s of each check gas period is used to calculate a mean value for each species. Figure 12 shows each instance of $N_2O$, $CO_2$, and CO check gas sampling, along with histograms for the difference from the known value. The time series show the last 5 s of each check gas period, along with a horizontal line indicating the known value of the air tank calibrated with the WMO standards as in Sect. 2.3. Note that the "check gas" and "calibration gas" cylinders were switched halfway through the campaign due to gas consumption, as reflected by the horizontal line. Looking at the difference of each check gas period from the known value, we find median offsets of 0.06 ppb, 0.06 ppm, and 0.03 ppb for $N_2O$, $CO_2$, and CO respectively, representative of possible bias between the flight system and the WMO scale. The $1\sigma$ values for the check gas points are 0.10 ppb, 0.30 ppm, and 1.62 ppb for $N_2O$, $CO_2$, and CO, representative of traceability of individual 1 s observations to the WMO scale. Table 1 summarizes the precision and accuracy for the four gases, though

we were unable to measure $H_2O$ traceability because we calibrated with dry tank air. We do report water vapor (and carbon dioxide) performance in comparison with the Picarro. Total instrument 1 s uncertainty is derived from summing in quadrature the $1\sigma$ accuracy to WMO, water vapor correction, and standard tank calibration uncertainty.

**Table 1.** Precision and accuracy for $N_2O$, $CO_2$, CO, and $H_2O$.

| | $N_2O$ (ppb) | $CO_2$ (ppm) | CO (ppb) | $H_2O$ (ppm) |
|---|---|---|---|---|
| $1\sigma$ Precision | 0.05 | 0.10 | 1.00 | 10 |
| Accuracy (median offset) | 0.06 | 0.06 | 0.03 | NA |
| $1\sigma$ comparison with Picarro | NA | 0.45 | NA | 340 |
| Accuracy ($1\sigma$ check gas) | 0.10 | 0.30 | 1.62 | NA |
| Water vapor correction | 0.023 | 0.076 | 0.75 | NA |
| WMO standard calibration | 0.26 | 0.11 | 0.71 | NA |
| Total $1\sigma$ uncertainty | 0.28 | 0.33 | 1.92 | NA |

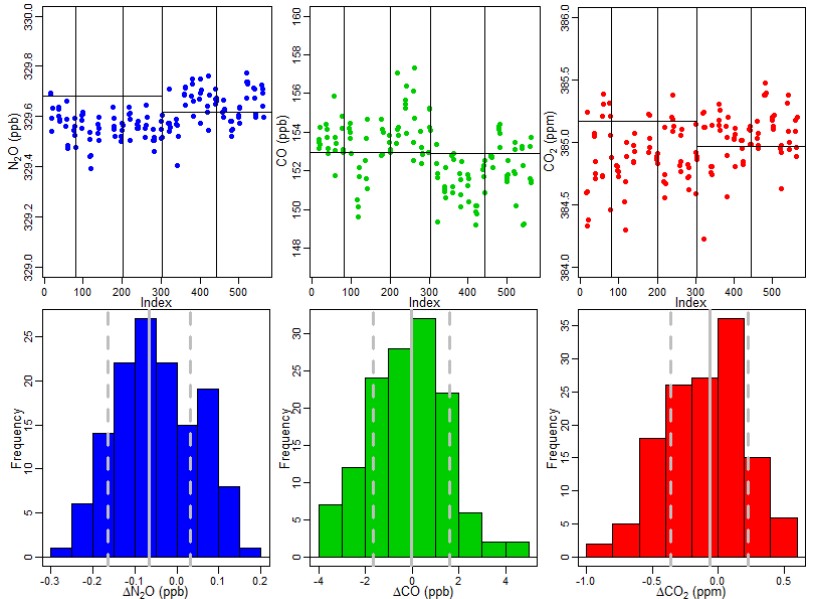

**Figure 12.** Top row: the last 5 s of each check gas period, black horizontal line indicating the value of the sampled gas traced to the WMO scale. Vertical lines separate the individual research flights. Bottom row: histograms of difference between known check gas value and last 5 s of measured check gas value, with solid gray lines indicating median and dashed lines showing $1\sigma$ uncertainty.

## 5 Applications

Continuous airborne $N_2O$ observations can be useful for quantifying fluxes and estimating emissions on a facility-to-regional scale. Mass balances techniques, which have been utilized to estimate emissions of other atmospheric gases as in Karion et al. (2013), Smith et al. (2015), Peischl et al. (2015), and Kort et al. (2016), could similarly be applied for $N_2O$. Figure 13 shows
the path flown during a research flight on May 6, 2017, with measured $N_2O$ mole fraction in color, white arrows indicating wind direction and speed, and blue and black arrows showing the direction of the flight route and the upwind and downwind transects. The downwind transect was flown at a mean altitude of 1515 msl, $1\sigma$ of 14 m, and the upwind transect at a mean altitude of 1501 msl, $1\sigma$ of 14m. The bottom right panel of the figure shows $N_2O$ from this flight as a function of latitude with the upwind and downwind transects in blue and black, while the top right panel shows the difference in $N_2O$ between
the downwind and upwind at each latitude. There is a distinct enhancement in the downwind transect relative to the upwind transect in the lower latitudes, from about 31.5° N to 32° N. This enhancement disappears at higher latitudes and the $N_2O$ measurement tracks well between upwind and downwind transects, even with a substantial latitudinal gradient. This flight illustrates the ability of this instrument to accurately measure small variations and link to local emissions (to the south) or larger scale gradients (to the north). Future analyses of this data can involve mass balance flux quantification and/or regional
model comparisons, both to quantify emissions and link to driving factors such as soil moisture or crop type.

  As a fast-response sensor, FCHAOS can also be used for point source quantifications, as first explained in Conley et al. (2017) and further analyzed in Mehrotra et al. (2017) and Vaughn et al. (2017). During FEAST, we circled several fertilizer plants with significant $N_2O$ emissions, and future analyses can leverage these observations to better quantify emissions from the large point sources.

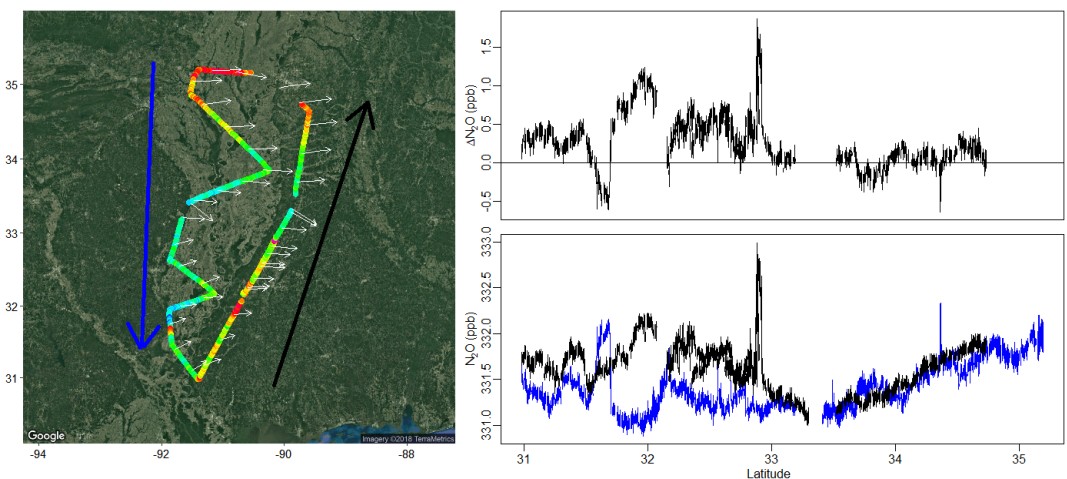

**Figure 13.** Left panel: flight path with $N_2O$ signal and wind direction (white arrows). Blue and black arrows show the direction of the planes route and indicate upwind and downwind transects. Bottom right panel: $N_2O$ signal as a function of latitude with upwind and downwind transects colored by blue and black, respectively. Top right panel: Difference in $N_2O$ between downwind and upwind transects as a function of latitude.

## 6 Conclusions

We present a continuous-wave, mid-IR laser spectrometer system that can measure continuous 1 Hz airborne mole fractions of $N_2O$, $CO_2$, CO, and $H_2O$. The commercial analyzer, when operated off-the-shelf, exhibits a dependence of $N_2O$, $CO_2$, and CO on cabin pressure. We correct for this artifact by employing an updated calibration procedure with mass flow control at a high flow rate enabling high-frequency, short-duration calibrations. While modern systems conventionally use pressure control and infrequent, long-duration zeros, our method expands on these previous approaches and opens up uses for the instrument in ways that have not yet been realized. We solve the inability of other systems to operate with large changes in cabin pressure by mitigating the cabin pressure effect while maintaining a ~90% duty cycle. In-flight $1\sigma$ precisions are estimated to be $\pm0.05$ ppb, $\pm0.10$ ppm, $\pm1.00$ ppb, and $\pm10$ ppm for $N_2O$, $CO_2$, CO, and $H_2O$, with total uncertainty in traceability estimated at 0.28 ppb, 0.33 ppm, and 1.92 ppb for $N_2O$, $CO_2$, and CO. We then validate our method by comparing FCHAOS data to $CO_2$ and $H_2O$ measurements from a flight-proven cavity ring-down spectrometer, seeing excellent agreement. This flight-proven system can provide key insights into $N_2O$ emissions processes by providing observational support for facility-quantification, for mass-balance flux estimates, and for inverse modeling. As presented, this system can be utilized for precise, accurate, continuous 1 Hz airborne observations of $N_2O$, $CO_2$, CO, and $H_2O$.

*Data availability.* Kort E.A., Gvakharia A., Smith M.L., Conley S. Airborne Data from the Fertilizer Emissions Airborne Study (FEAST). Nitrous Oxide, Carbon Dioxide, Carbon Monoxide, Methane, Ozone, Water Vapor, and meteorological variables over the Mississippi River Valley [Data set]. University of Michigan Deep Blue Data Repository. https://doi.org/10.7302/Z2XK8CRG

*Competing interests.* The authors declare they have no competing interests.

*Acknowledgements.* We thank Scientific Aviation pilots for their efforts in helping collect this data and Aerodyne Research Inc. for useful discussions. This material is based partly upon work supported by the National Science Foundation under Grant No. 1650682.

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
