# Peer review of "Testing and evaluation of a new airborne system for continuous N2O, CO2, CO, and H2O measurements: the Frequent Calibration High-performance Airborne Observation System (FCHAOS)"

_Atmospheric Measurement Techniques, 2018_

## Referee Comment (RC1) · J.R. Pitt (Referee) · 19 Aug 2018

It has now become well-known in the trace gas measurement community that analysers employing Tunable Infrared Laser Direct Absorption Spectroscopy (TILDAS) techniques on aircraft can exhibit a strong sensitivity to changes in cabin pressure. This is currently a major limitation to the utility of these instruments for airborne sampling, as operators either have to accept large altitude-dependent biases in their final dataset or

very low duty cycles (as the instrument must be recalibrated at each altitude). Many important trace gas species (e.g. N2O and C2H6) have much stronger absorption lines in the mid-infrared than the near-IR; whilst cavity-based measurement techniques for these species have developed significantly in recent years, TILDAS is still the most commonly used technique for measuring many trace gases in the mid-IR. Therefore improving the accuracy and duty cycle of TILDAS instruments during aircraft sampling is important if we are to improve our understanding of key greenhouse and pollutant gases.

This paper presents a novel calibration strategy to tackle this issue, resulting in greatly reduced altitude-dependent biases with a 90% duty cycle. The switch to controlling mass flow instead of pressure is a clever idea, removing issues associated with pressure instability during the sample-calibration transition. The method employed here will certainly be of great interest to anyone currently operating TILDAS analysers on aircraft, but it also provides the potential for reducing biases and/or increasing the duty cycle for other airborne instrumentation. I recommend its publication in AMT, but I have the following suggestions for minor revisions.

The water broadening correction is mentioned at the end of section 2.1. Did you determine the water broadening to air broadening ratio yourself experimentally? If so perhaps state this explicitly because at the moment it makes it seem like TDLWintel does this automatically without user intervention (unless Aerodyne tested your instrument before sending it to you – in which case mention this because as far as I'm aware it isn't what they don't usually do). Assuming you determined the coefficients yourself, could you add a brief outline of the general approach taken (e.g. H2O injection/dew point generator/etc...) and the uncertainties associated with it please? The uncertainty associated with this water vapour correction is often on the same order as the other uncertainties so it's important to consider it.

Section 2.3 is presented in a rather confusing way. I think this is largely because the use of two in-flight calibration gases is introduced here, but the fact that one of them

is used as a check gas is not mentioned until section 4.1. It wasn't until I got to this point much later in the text that I fully understood what was going on (e.g. why the long-term drift in instrument slope needed quantifying in the lab) – it would be much better if it was explicitly stated in section 2.3 that a single-point calibration strategy with an additional check gas was used in-flight. Additionally, the term "linearity" is used in a way here that isn't intuitive to me. I would stick to using the word "slope" throughout (or "gain" if an alternative is needed), as what is being tested here is the extent to which the instrument slope drifts with time. In my mind instrument linearity is the extent to which the linear fit used here is applicable, not whether the coefficients are drifting with time. To assess linearity three cylinders with different mole fractions are therefore needed. This hasn't been done here, but these instruments are known to have a good linear response so it's probably safe to assume non-linearity is a small component of the uncertainty budget.

Section 3.2: having had many discussions with Aerodyne about this following on from our 2014/15 campaigns, I am fairly certain that the main source of the large gradients in mole fraction you see during profiles is indeed an optical fringe (or possibly multiple fringes), activated by the change in cabin pressure. However we do also see artefacts in the measurements associated with aircraft acceleration as well – for us these manifest themselves as much smaller-timescale features which would not be captured by the calibration strategy here.

Section 4: our experience is that there is significant flight-to-flight variability in the cabin pressure artefact, making it impossible to apply a single correction throughout a campaign. This is because, while the FSR of the fringe at ground level stays constant, its initial phase is variable and unpredictable. We did experiment with doing two deep profiles at the beginning and end of a flight on tank air to try and calibrate out this effect, but the drift during the flight resulted in a large uncertainty and we abandoned this approach. Essentially I'm not convinced that this effect is repeatable enough to take data from a single null flight and use it to correct other flights, so I'd probably

remove that paragraph (unless you have evidence to the contrary). The method you've developed here seems far superior to any that could be developed from the null test results.

I'd be interested to know more about the suspected contamination which has resulted in the 0.6 ppm $CO_2$ offset between the Picarro and the FCHAOS systems. Have you been able to identify which of the systems the contamination is associated with? Reading the manuscript I initially assumed that it was the FCHAOS regulator/tubing that was under suspicion, but it would be good to make this explicit, or if it is still unknown which instrument was contaminated to state that. Was there any change to the setup in future campaigns where this offset was not observed? The regulator and tubing used for the FCHAOS here are pretty standard, so if there are contamination effects associated with either it would be good to know about them! Or is the theory that the cylinder itself became contaminated (e.g. due to a mistake during regulator flushing)? Surely in that case you would also expect to see the offset in the check cylinder data in Figure 8? In the first half of the campaign (before the cylinder switch) there may be a sign of a negative bias in $CO_2$, but there doesn't appear to be a corresponding positive bias in the second half, so this doesn't really tally with a single cylinder contamination. Also in that case you'd expect the Picarro bias only to be present in one half of the campaign but I can't see any evidence of this in Figure 7. If you haven't made any further progress in diagnosing this then no worries, but it would be good to include any extra details you do have.

The fact you don't see the same effect on the $H_2O$ measurements is very interesting, but I can't quite see this from the plots included here. Could you add another column in Fig. 4 showing $H_2O$ please? I know the tanks were dry but you are assuming (in my view reasonably) that the artefact is a simple offset shift so it shouldn't matter what the absolute value of the $H_2O$ mole fraction is – even if it is completely dry I'd expect the fringing to affect this zero-offset reading. I don't doubt your word here, but the explanations offered as to why the fringe would affect $H_2O$ less don't really make sense

to me either so I could do with a bit more detail on these. Surely the fringe amplitude will increase with laser intensity, if anything making the problem worse? The relevant signal-to-noise here is the strength of the absorption peak (not the laser intensity) relative to the fringe amplitude. If the H2O line in question is the one at ~2227.5 cm-1 then this tends to be a weak feature relative to the N2O and CO2 lines, so again I would have thought that H2O would be more affected by the fringe. I'm also not sure why having a line frequency of ~2227.5 cm-1 compared to the CO2 line at ~2227.6 cm-1 (for instance) would reduce the fringe interference. It is definitely true that a very wide absorption feature would suffer less from a fringe with a small FSR, but I don't think the H2O line is wider than the other peaks here? Sorry to labour the point on this, but the fact that you don't observe the fringe effect on the H2O measurement could be a really useful piece of information in trying to further mitigate this issue, so I'm keen to better establish the cause of it.

Section 5: Could you put details of the altitude and variability in altitude during the runs in here please? I assume it was essentially performed at a single level but if so it would be good to be clear about this just so the reader knows there are no vertical gradients convolved in here.

Specific points:

P3 L2 – Minor point but the LGR FGGA in O'Shea et al. is a near-IR instrument

P3 L26 – Typo: missing space

P5 L18 – "...within our 1 Hz precision..."

P7 L9 – "...at the same flow-rate..."

P7 L12 – What interpolation technique was used?

P8 L8 – "...within our 1 Hz precision."

---

## Referee Comment (RC2) · Anonymous Referee #2 · 21 Aug 2018

Review of Gvakharia et al.:

This work presents airborne data of an in-situ QCL absorption spectrometer measuring greenhouse gases N2O, CO2, CO, and H2O with a commercial Aerodyne spectrometer. Such instruments tend to show strong drifts due to changing pressure and/or temperature inside the aircraft cabin. This holds particular during ascent and descent and for unpressurized platforms.

[Figure]

To reduce the effect of the drift, the authors apply a calibration system, which is new according to the authors - the frequent calibration high performance airborne observation system (FCHAOS). Basically the absorption cell of the IR-spectrometer is frequently flushed with a high flow of calibration cylinders with ambient mixing ratios of the target gases tracable to the NOAA WMO scale. The authors apply a duty cycle of 120 s and purge the cell for 10 s with additional 5 s latency before measuring. The output frequency is 1 Hz. The authors show, that by applying this calibration procedure the effect of the drift is accounted for. In-flight comparisons with a PICARRO CRDS system confirms this. The correction is shown for N2O data during a research flight and demonstrates the effect of the procedure.

The paper is well written and documents the calibration procedure allowing to reduce instrumental drift particularly during ascent and descent. I fully acknowledge a clear documentation of instrumental performance and data processing. However, I can't see the novelty of the approach. Fast flow and frequent short calibration with subsequent linear drift correction is basically, what has been applied here. Note, that 1.5 slpm are not novel (e.g. Korrmann et al., 2005 used 1-1.5 slpm at 56 hPa, cell < 0.5 l) as well as linear drift correction is standard.

If the authors could show, that the regulation of mass flow (MFC) upstream the cell (and downstream the cal switch valve) is the key to guarantee short calibration times by reducing pressure pulses (as suggested in the conclusions) I would see a potential new aspect. For this they should provide e.g. comparisons between pressure and flow controlled approaches (see below). It is not shown, why a pressure controlled system should not have the same performance.

As the paper currently stands, it is a well documented calibration procedure of a commercial instrument with standard methods. Therefore I don't see the paper in AMT in its current form.

Main point:

[Figure]

If the use of an MFC is the key novelty this should be clearly documented in the analysis. The current Fig.1. and the text states, that three-way solenoid valves are used (p.4, l.16/17). In case of a calibration I expect a direct connection between the pressure transducer (calibration tank) and the MFC controlling the cell flow and thus a pressure pulse. The inlet line is probably closed during calibration. In case of switching from ambient to calibration I still expect a short pressure pulse perturbing MFC and cell pressure. This will probably stabilize after a few seconds since the MFC limits the flow, but I do not see the advantage or novelty over a calibration using overflow of calibration gas by flushing the inlet at ambient pressure, which has been applied since years to GHG measurements by TDLAS (or QCLAS).

Note, that many QCLAS or TDLAS systems often are calibrated by flushing the inlet line with higher flow rates than the cell flow. The calibration gas tube is directly connected to the inlet and thus ambient pressure solely via a t-connector in the inlet line. Calibration gas is just switched via an open/close valve. This ensures a minimum pressure perturbation of the cell due to the open connection of the calibration line to the inlet.

This has been established over a long time (e.g. Wienhold et al., 1998) and a potential advantage - if existing - via the proposed procedure in Gvakharia et al., should be documented.

p.7. l.10-20: Would be good to see a highly resolved single calibration signal with individual data points and the cell pressure for a ground test and in-flight conditions at lower ambient pressure.

p.13, l.5: How do the respective Allan variance plots look like for the Nulltest? How do they compare to a lab test?

Fig.4: y-Axis: mixing ratio instead of concentration (also check the main text).

References: Wienhold et al.: TRISTAR – a tracer in situ TDLAS for atmospheric research, Applied Optics B, 67, 1998.

Kormann et al., QUALITAS: A mid-infrared spectrometer for sensitive trace gas measurements based on quantum cascade lasers in CW operation, Rev. Sci. Instr., 76, 2005.

---

## Referee Comment (RC3) · Anonymous Referee #1 · 26 Aug 2018

The paper describes an airborne Tunable Infrared Laser Direct Absorption Spectroscopy (TILDAS) system for airborne atmospheric trace gas measurements. The focus is on a novel method to address cabin pressure induced changes in the measured mole fraction. The paper is well written, and fits well within the scope of AMT. However, a few issues listed below should be addressed before the paper can be recommended for publication.

General comments:

The potential impact of water vapour on the derived dry air mole fractions should be more elaborated. E.g. Pitt et al. (2016) mention a lack of long-term stability in the retrieval of H2O mole fractions using a similar instrument. Has the wet-dry correction been tested, and if so, what was the setup used for this? Also the difference of the Picarro G2301-f and FCHAOS water vapour measurements shows a standard deviation of 0.034%, which would correspond to uncertainties of 0.136 ppm CO2 (at 400 ppm) just due to dilution by H2O alone, more than the claimed precision of 0.1 ppm.

Specific comments:

Fig. 1: In addition to the calibration tanks, there should also be a symbol for e.g. a pressure regulator of valve included.

Pg 4 Line 14: please clarify if there is excess flow escaping backward through the inlet when calibrating, or if calibration is performed by replacing the sample gas (solenoid valve closed to ambient, only open to filter/MFC). In the latter case I would expect slightly larger fluctuations in pressure within the inlet and filter,

---

## Author Comment (AC1) · 17 Oct 2018

Thank you for the thorough review and suggestions. We have responded to each comment in-line below and updated the text—adding more text on key issues raised and including more figures to support the discussion. We believe the manuscript is now much improved.

**J. R. Pitt:**

It has now become well-known in the trace gas measurement community that analysers employing Tunable Infrared Laser Direct Absorption Spectroscopy (TILDAS) techniques on aircraft can exhibit a strong sensitivity to changes in cabin pressure. This is currently a major limitation to the utility of these instruments for airborne sampling, as operators either have to accept large altitude-dependent biases in their final dataset or very low duty cycles (as the instrument must be recalibrated at each altitude). Many important trace gas species (e.g. N2O and C2H6) have much stronger absorption lines in the mid-infrared than the near-IR; whilst cavity-based measurement techniques for these species have developed significantly in recent years, TILDAS is still the most commonly used technique for measuring many trace gases in the mid-IR. Therefore improving the accuracy and duty cycle of TILDAS instruments during aircraft sampling is important if we are to improve our understanding of key greenhouse and pollutant gases.

This paper presents a novel calibration strategy to tackle this issue, resulting in greatly reduced altitude-dependent biases with a 90% duty cycle. The switch to controlling mass flow instead of pressure is a clever idea, removing issues associated with pressure instability during the sample-calibration transition. The method employed here will certainly be of great interest to anyone currently operating TILDAS analysers on aircraft, but it also provides the potential for reducing biases and/or increasing the duty cycle for other airborne instrumentation. I recommend its publication in AMT, but I have the following suggestions for minor revisions.

The water broadening correction is mentioned at the end of section 2.1. Did you determine the water broadening to air broadening ratio yourself experimentally? If so perhaps state this explicitly because at the moment it makes it seem like TDLWintel does this automatically without user intervention (unless Aerodyne tested your instrument before sending it to you – in which case mention this because as far as I'm aware it isn't what they don't usually do). Assuming you determined the coefficients yourself, could you add a brief outline of the general approach taken (e.g. H2O injection/dew point generator/etc. . .) and the uncertainties associated with it please? The uncertainty associated with this water vapour correction is often on the same order as the other uncertainties so it's important to consider it.

Thank you for raising this point – this is of great importance to accurate trace gas observations and we have added a section discussing this. We now describe the water broadening coefficients, our determination of the values, and the associated uncertainties, and we have added a figure showing the effect of the correction as a function of water vapor. We used a moist filter similar to Lebegue et al. 2016, and sampled a mixture of wet and dry tank air which we then reanalyzed to find the proper coefficients. Table 1 has also been updated to include these uncertainties. We also note that in adding this discussion we realized the proper water broadening coefficients were not consistently applied to the data presented initially. We have corrected this oversight for this dataset in all the data and figures presented in the revision.

Section 2.3 is presented in a rather confusing way. I think this is largely because the use of two in-flight calibration gases is introduced here, but the fact that one of them is used as a check gas is not mentioned until section 4.1. It wasn't until I got to this point much later in the text that I fully understood what was going on (e.g. why the long-term drift in instrument slope needed quantifying in the lab) – it would be much better if it was explicitly stated in section 2.3 that a single-point calibration strategy with an additional check gas was used in-flight. Additionally, the term "linearity" is used in a way here that isn't intuitive to me. I would stick to using the word "slope" throughout (or "gain" if an alternative is needed), as what is being tested here is the extent to which the linear fit used here is applicable, not whether the coefficients are drifting with time. To assess linearity three cylinders with different mole fractions are therefore needed. This hasn't been done here, but these instruments are known to have a good linear response so it's probably safe to assume non-linearity is a small component of the uncertainty budget.

**We have updated Section 2.3 to clarify this point as suggested. We have also changed the use of "linearity" to the more appropriate "slope" throughout.**

Section 3.2: having had many discussions with Aerodyne about this following on from our 2014/15 campaigns, I am fairly certain that the main source of the large gradients in mole fraction you see during profiles is indeed an optical fringe (or possibly multiple fringes), activated by the change in cabin pressure. However we do also see artefacts in the measurements

associated with aircraft acceleration as well – for us these manifest themselves as much smaller-timescale features which would not be captured by the calibration strategy here.

Optical fringes are likely the dominant component. Acceleration-induced artifacts can have very short duration (g-force), and not be well-captured by the method here. They can also have a longer duration (e.g. large engine pull for the duration of a climb possibly impacting instrument via electrical feedbacks), and these would be captured. As shown in the paper, longer-duration features dominate our artifact and thus are the focus of improvements.

**We have added clarification to the end of first paragraph in section 3.2 "Artifacts that occur on shorter time-frames, such as induced by a short duration turbulence event, will not be corrected with this method."**

Section 4: our experience is that there is significant flight-to-flight variability in the cabin pressure artefact, making it impossible to apply a single correction throughout a campaign. This is because, while the FSR of the fringe at ground level stays constant, its initial phase is variable and unpredictable. We did experiment with doing two deep profiles at the beginning and end of a flight on tank air to try and calibrate out this effect, but the drift during the flight resulted in a large uncertainty and we abandoned this approach. Essentially I'm not convinced that this effect is repeatable enough to take data from a single null flight and use it to correct other flights, so I'd probably remove that paragraph (unless you have evidence to the contrary). The method you've developed here seems far superior to any that could be developed from the null test results.

**We agree that our method is an improvement on attempting a single cabin pressure artifact correction. We've modified the text in Section 4 as outlined below to clarify.**

"Given the repeatable, smooth nature of the cabin pressure artifact, it *would seem* possible to use just the cabin pressure data to empirically correct for the artifact, without running frequent calibrations. This method *would not* account for long-term spectral drift however or traceability, *and relies on the assumption that the cabin pressure artifact will be stable and repeatable. These weaknesses compromise such an approach.*"

I'd be interested to know more about the suspected contamination which has resulted in the 0.6 ppm CO2 offset between the Picarro and the FCHAOS systems. Have you been able to identify which of the systems the contamination is associated with? Reading the manuscript I initially assumed that it was the FCHAOS regulator/tubing that was under suspicion, but it would be good to make this explicit, or if it is still unknown which instrument was contaminated to state that. Was there any change to the setup in future campaigns where this offset was not observed? The regulator and tubing used for the FCHAOS here are pretty standard, so if there are contamination effects associated with either it would be good to know about them! Or is the theory that the cylinder itself became contaminated (e.g. due to a mistake during regulator flushing)? Surely in that case you would also expect to see the offset in the check cylinder data in Figure 8? In the first half of the campaign (before the cylinder switch) there may be a sign of a negative bias in CO2, but there doesn't appear to be a corresponding positive bias in the second half, so this doesn't really tally with a single cylinder contamination. Also in that case you'd

expect the Picarro bias only to be present in one half of the campaign but I can't see any evidence of this in Figure 7. If you haven't made any further progress in diagnosing this then no worries, but it would be good to include any extra details you do have.

**We appreciate your interest in this cause. We have gone back through in further detail trying to isolate a potential cause, but still are left with a somewhat unsatisfactory suspicion of contamination. We have modified the text in Section 4 slightly in accordance with this. We also add the discussion here to further highlight the importance of the water vapor correction you raised earlier, noting that residual water vapor errors (in either the FCHAOS or Picarro systems) could contribution to perceived biases.**

The fact you don't see the same effect on the H2O measurements is very interesting, but I can't quite see this from the plots included here. Could you add another column in Fig. 4 showing H2O please? I know the tanks were dry but you are assuming (in my view reasonably) that the artefact is a simple offset shift so it shouldn't matter what the absolute value of the H2O mole fraction is – even if it is completely dry I'd expect the fringing to affect this zero-offset reading. I don't doubt your word here, but the explanations offered as to why the fringe would affect H2O less don't really make sense to me either so I could do with a bit more detail on these. Surely the fringe amplitude will increase with laser intensity, if anything making the problem worse? The relevant signal-to-noise here is the strength of the absorption peak (not the laser intensity) relative to the fringe amplitude. If the H2O line in question is the one at  $\sim$ 2227.5 cm-1 then this tends to be a weak feature relative to the N2O and CO2 lines, so again I would have thought that H2O would be more affected by the fringe. I'm also not sure why having a line frequency of ~2227.5 cm-1 compared to the CO2 line at ~2227.6 cm-1 (for instance) would reduce the fringe interference. It is definitely true that a very wide absorption feature would suffer less from a fringe with a small FSR, but I don't think the H2O line is wider than the other peaks here? Sorry to labour the point on this, but the fact that you don't observe the fringe effect on the H2O measurement could be a really useful piece of information in trying to further mitigate this issue, so I'm keen to better establish the cause of it.

Thank you for digging into this question further. We have updated Section 4 when discussing comparisons between FCHAOS and the Picarro, and added the figure below to show the water vapor during the null test profiles as well as difference between FCHAOS and Picarro measured water vapor as a function of altitude. The long equilibration time for water vapor makes it difficult for us to see an altitude artifact if it is present. One null test profile suggests there may be a ~60 ppm water vapor vertical sensitivity, but it is difficult to clearly establish. In the comparison with the Picarro we see no evidence of any vertical dependency of water vapor, though a dependency of 10s of ppm would be hidden within the respective instruments noise. We have updated the text to include this more nuanced discussion.

---

## Author Comment (AC3) · 17 Oct 2018

**We thank Referee #1 for the review and suggestions. We have addressed the comments line-by-line below and updated the manuscript as needed, and feel this has improved the paper.**

**Anonymous Referee #1:**

The paper describes an airborne Tunable Infrared Laser Direct Absorption Spectroscopy (TILDAS) system for airborne atmospheric trace gas measurements. The focus is on a novel method to address cabin pressure induced changes in the measured mole fraction. The paper is well written, and fits well within the scope of AMT. However, a few issues listed below should be addressed before the paper can be recommended for publication.

**Thank you for the useful feedback. We respond inline below.**

**General comments:**

The potential impact of water vapour on the derived dry air mole fractions should be more elaborated. E.g. Pitt et al. (2016) mention a lack of long-term stability in the retrieval of H2O mole fractions using a similar instrument. Has the wet-dry correction been tested, and if so, what was the setup used for this? Also the difference of the Picarro G2301-f and FCHAOS water vapour measurements shows a standard deviation of 0.034%, which would correspond to uncertainties of 0.136 ppm CO2 (at 400 ppm) just due to dilution by H2O alone, more than the claimed precision of 0.1 ppm.

We address the water vapor correction in response to Dr. Pitt's review, so please refer to that above for more detail. We have tested the wet-dry correction and included information on the setup in Section 2.4. We have added new text and figures, regarding both the uncertainties due to water and any altitude-dependent sensitivity. This is an important issue so we have expanded our manuscript accordingly.

The 1 $\sigma$  difference of 340 ppm in water vapor between the two instruments would indeed cause a dilution uncertainty larger than the 1s precision of the instrument. We note that the precision is not where such an uncertainty should be described; instead potential water vapor correction errors impact the accuracy of the instrument and traceability to a known scale. Using the traceability standard we find an empirical check gas instrument accuracy of 0.30 ppm for  $CO_2$  (in Table 1). The comparison with the Picarro shows variance of differences between the analyzers of 0.45 ppm for 1 s data. This slightly elevated value compared to the check gas approach may be related to the dilution, but may simply be variance in the Picarro. We have added this information to the manuscript in the Table to help clarify, and now have a total uncertainty value that accounts for the various sources of uncertainty, including the water vapor uncertainty impact.

**Specific comments:**

Fig. 1: In addition to the calibration tanks, there should also be a symbol for e.g. a pressure regulator of valve included.

**Thank you for catching the omission, we have updated Figure 1.**

Pg 4 Line 14: please clarify if there is excess flow escaping backward through the inlet when calibrating, or if calibration is performed by replacing the sample gas (solenoid valve closed to ambient, only open to filter/MFC). In the latter case I would expect slightly larger fluctuations in pressure within the inlet and filter,

There is no excess flow, the calibration does indeed switch directly from the inlet to the tanks with the solenoid valves to preventing potential contamination of other inlets and also not consume excess calibration gas. We have added a figure in Section 2.2 that shows the pressure variation when switching to a calibration gas in our system and clarified the text to explain our setup (replacement, no excess flow).

---

## Author Comment (AC2)

**We thank Referee #2 for the suggestions and concerns. We have addressed the comments specifically below, and have updated the manuscript accordingly. We do disagree with the perspective that this paper only presents a calibration procedure (that is not novel to this reviewer) and thus is not worthy of publication in AMT, and outline our perspective below. As some of this concern may have arisen due to a lack of clarity in our writing, we have made changes to the manuscript both to clarify that our approach is an extension of traditional calibration methods and highlight the uniqueness of the approach. We have also added figures as requested more clearly illustrated to impact of mass flow control and the appearance of our calibrations. Overall we feel these changes have improved the manuscript.**

**Anonymous Referee #2:**

This work presents airborne data of an in-situ QCL absorption spectrometer measuring greenhouse gases N2O, CO2, CO, and H2O with a commercial Aerodyne spectrometer. Such instruments tend to show strong drifts due to changing pressure and/or temperature inside the aircraft cabin. This holds particular during ascent and descent and for unpressurized platforms.

To reduce the effect of the drift, the authors apply a calibration system, which is new according to the authors - the frequent calibration high performance airborne observation system (FCHAOS). Basically the absorption cell of the IR-spectrometer is frequently flushed with a high flow of calibration cylinders with ambient mixing ratios of the target gases tracable to the NOAA WMO scale. The authors apply a duty cycle of 120 s and purge the cell for 10 s with additional 5 s latency before measuring. The output frequency is 1 Hz. The authors show, that by applying this calibration procedure the effect of the drift is accounted for. In-flight comparisons with a PICARRO CRDS system confirms this. The correction is shown for N2O data during a research flight and demonstrates the effect of the procedure.

The paper is well written and documents the calibration procedure allowing to reduce instrumental drift particularly during ascent and descent. I fully acknowledge a clear documentation of instrumental performance and data processing. However, I can't see the novelty of the approach. Fast flow and frequent short calibration with subsequent linear drift correction is basically, what has been applied here. Note, that 1.5 slpm are not novel (e.g. Korrmann et al., 2005 used 1-1.5 slpm at 56 hPa, cell < 0.5 l) as well as linear drift correction is standard.

If the authors could show, that the regulation of mass flow (MFC) upstream the cell (and downstream the cal switch valve) is the key to guarantee short calibration times by reducing pressure pulses (as suggested in the conclusions) I would see a potential new aspect. For this they should provide e.g. comparisons between pressure and flow controlled approaches (see below). It is not shown, why a pressure controlled system should not have the same performance.

As the paper currently stands, it is a well documented calibration procedure of a commercial instrument with standard methods. Therefore I don't see the paper in AMT in its current form.

**We respectfully disagree with the perspective that this paper only presents a calibration procedure (that is not novel to this reviewer) and thus is not worthy of publication in AMT. This criticism is founded on two underlying perspectives: 1) That the only new value of this paper is the presentation of a new calibration method & 2) the calibration method is not novel. We disagree with both perspectives. Firstly, the paper is not solely about a calibration method. This paper is the first presentation of this flight system and shows extensive validation with in-flight null tests and direct comparison with a Picarro. Even if the calibration approach was not novel, reporting the instrument performance and traceability with validation in such detail warrants publication on its own in AMT, and in fact is necessary for the community to have confidence in the data reported from this flight system, particularly considering the most similar flight system published cannot collect data during vertical profiles (Pitt et al. 2016). This is consistent with AMT standards and the expectation that papers "comprise the development, intercomparison, and validation of measurement instruments and techniques …" as evidenced by similar publications focused on intercomparison and validation (for example Santoni et al. 2014 & Pitt et al. 2016).**

**Regarding the calibration approach, it certainly can be considered a natural extension of previous calibration approaches. In spite of the apparent triviality of the approach, the combination of high flow rate, mass flow**

control and high frequency, short duration calibrations with near-ambient concentration cal gas has never been applied to other modern GHG systems. The most recent, state-of-the-science instrument paper on a flight QCLS N$_2$O system concluded they couldn't use data during vertical profiles (Pitt et al. 2016)—an important limitation for a flight instrument. While our approach may seem simple, we are able to achieve unprecedented in-flight performance in the face of dynamic environmental variables (cabin pressure, etc.), thus achieving a better duty cycle and more robust performance than any published flight N$_2$O system. We feel the overall approach is novel (and we note the other reviewers do as well), but even if the calibration method is felt to be simple, the extensive validation presented in the manuscript is new and necessary to document this systems performance.

Some of this concern may have arisen from lack of clarity in our writing. In response, we have made some changes to clarify that this is an extension of traditional methods and pinpoint the uniqueness of this approach (see abstract, intro, conclusion).

Also as requested, we have added a figure and discussion on the importance of the mass flow control approach (contrasting with the p-control setup) to more clearly illustrate some of the different elements/novelty of the setup. This is outlined more below.

Main point:

If the use of an MFC is the key novelty this should be clearly documented in the analysis. The current Fig.1. and the text states, that three-way solenoid valves are used (p.4, l.16/17). In case of a calibration I expect a direct connection between the pressure transducer (calibration tank) and the MFC controlling the cell flow and thus a pressure pulse. The inlet line is probably closed during calibration. In case of switching from ambient to calibration I still expect a short pressure pulse perturbing MFC and cell pressure. This will probably stabilize after a few seconds since the MFC limits the flow, but I do not see the advantage or novelty over a calibration using overflow of calibration gas by flushing the inlet at ambient pressure, which has been applied since years to GHG measurements by TDLAS (or QCLAS).

Note, that many QCLAS or TDLAS systems often are calibrated by flushing the inlet line with higher flow rates than the cell flow. The calibration gas tube is directly connected to the inlet and thus ambient pressure solely via a t-connector in the inlet line. Calibration gas is just switched via an open/close valve. This ensures a minimum pressure perturbation of the cell due to the open connection of the calibration line to the inlet.

This has been established over a long time (e.g. Wienhold et al., 1998) and a potential advantage - if existing - via the proposed procedure in Gvakharia et al., should be documented.

**Thank you for bringing this issue up. We are familiar with the excess flow, pressure control setup, and have flown such a system many times. We note one disadvantage of the excess flow setup is that it can lead to contamination of other instrument sampling for nearby inlets and thus is often not preferred (with a second weakness being larger cal gas consumption). Even with the excess flow setup, there will invariably be some pressure fluctuations in the cell when switching to calibration gas. Depending on the specifics of the pressure control setup, to achieve stable pressure control over the entire dynamic range sampled by the aircraft makes it a challenge to prevent any pressure blips when switching to calibration gas, though it may be possible. We show below here an example of a calibration with a pressure control setup (including excess flow) compared with the mass flow control. For our system, we have peak-to-peak fluctuations on the order of 0.8 Torr occur with the pressure control setup and transient pressure fluctuations that do not stabilize within 10s. With mass flow control, they are reduced to ~0.04 Torr, an order of magnitude smaller, and stabilize in much shorter times. We have added this figure and some related discussion to the manuscript in Section 2.2.**

[Figure]

p.7. l.10-20: Would be good to see a highly resolved single calibration signal with individual data points and the cell pressure for a ground test and in-flight conditions at lower ambient pressure.

**The plot below shows the calibration signal in flight, with ambient pressure around 730 mb. Similar to what was seen in the pressure plot above, the fluctuation in pressure is minimal when the valve changes. We have also added this figure to the manuscript in Section 3.2.**

[Figure]

p.13, l.5: How do the respective Allan variance plots look like for the Nulltest? How do they compare to a lab test?

**The top set of plots shows Allan variance for the 04/26 null test. The bottom set shows Allan variance plots for when gas was sampled on the ground (note tanks are dry so there is no $H_2O$). As is illustrated, the in-flight performance closely matches the ground performance, except noise is increased by a factor of 2. We have added text accordingly to make this point in Section 4.1.**

[Figure]

Fig.4: y-Axis: mixing ratio instead of concentration (also check the main text).

**Thank you for the suggestion, we have updated Figure 4 as well as Figure 2 and checked the consistency of the main text.**